# Barriers and facilitators of physical activity in knee and hip osteoarthritis: a systematic review of qualitative evidence

Archontissa M Kanavaki,[1,2] Alison Rushton,[1,2,3] Nikolaos Efstathiou,[4,8] Asma Alrushud,[1,5] Rainer Klocke,[6] Abhishek Abhishek,[7] Joan L Duda[1,2]

For numbered affiliations see end of article.

**Correspondence to**
Archontissa M Kanavaki;
amk377@bham.ac.uk

## ABSTRACT

Physical activity (PA), including engagement in structured exercise, has a key role in the management of hip and knee osteoarthritis (OA). However, maintaining a physically active lifestyle is a challenge for people with OA. PA determinants in this population need to be understood better so that they can be optimised by public health or healthcare interventions and social policy changes.

**Objectives** The primary aim of this study is to conduct a systematic review of the existing qualitative evidence on barriers and facilitators of PA for patients with hip or knee OA. Secondary objective is to explore differences in barriers and facilitators between (1) lifestyle PA and exercise and (2) PA uptake and maintenance.

**Methods** Medline, Embase, Web of Science, Cumulative Index to Nursing and Allied Health Literature, SPORTDiscus, Scopus, Grey literature and qualitative journals were searched. Critical Appraisal Skills Programme—Qualitative checklist and Lincoln and Guba's criteria were used for quality appraisal. Thematic synthesis was applied.

**Findings** Ten studies were included, seven focusing on exercise regimes, three on overall PA. The findings showed a good fit with the biopsychosocial model of health. Aiming at symptom relief and mobility, positive exercise experiences and beliefs, knowledge, a 'keep going' attitude, adjusting and prioritising PA, having healthcare professionals' and social support emerged as PA facilitators. Pain and physical limitations; non-positive PA experiences, beliefs and information; OA-related distress; a resigned attitude; lack of motivation, behavioural regulation, professional support and negative social comparison with coexercisers were PA barriers. All themes were supported by high and medium quality studies. Paucity of data did not allow for the secondary objectives to be explored.

**Conclusion** Our findings reveal a complex interplay among physical, personal including psychological and social-environmental factors corresponding to the facilitation and hindrance of PA, particularly exercise, engagement. Further research on the efficacy of individualised patient education, psychological interventions or social policy change to promote exercise engagement and lifestyle PA in individuals with lower limb OA is required.

**Trial registration number** CRD42016030024.

## Strengths and limitations of this study

► This systematic review is the first to identify, appraise and synthesise the existing qualitative research on barriers and facilitators to physical activity (PA) in knee and hip osteoarthritis.
► Rigorous methods have been applied, informed by the Centre for Reviews and Dissemination and Cochrane Qualitative Research Methods Group guidelines and reported according to the Preferred Reporting Items for Systematic Reviews and Meta-Analyses and Enhancing Transparency in Reporting the Synthesis of Qualitative Research statements.
► The majority of the included studies (7/10) focused on exercise barriers and facilitators; therefore, barriers and facilitators of more general lifestyle PA might not be fully captured.
► Papers written in English-language only were included.

Osteoarthritis (OA) is the the most common joint disease and main cause of disability in older adults.[1] OA management focuses on analgesia and non-pharmacological modalities such as exercise and weight loss.[2] Exercise, that is, structured and purposeful physical activity (PA),[3] reduces pain and improves function in people with knee or hip OA.[4–9] However, despite the positive effects on symptoms, exercise interventions do not promote sustained behaviour change.[10 11] Just like exercise, PA associates with better physical function[12–14] and even modest increase in PA (from sedentary to light intensity PA) improves arthritis pain.[15] At the population level, it is simpler to promote PA in people with painful OA for example, via radio and television, than promoting exercise as that will require a greater behaviour change and may need continued support of trained physiotherapists. However, existing evidence suggests that people with lower limb OA have such low PA levels that they

gain no health benefits from it.[16–18] Thus, there is a need to understand the determinants of reduced PA in people with symptomatic OA so that these can be optimised to promote PA.

The disease-specific determinants of PA in those with lower limb OA, for example, symptom severity and physical function[19–23] are relatively well understood, but the psychological, social and environmental determinants of PA in OA have not been adequately examined.[21 22] Understanding these factors is of great importance as pain makes PA an aversive experience leading to activity avoidance[24–27] and pain is influenced by psychological and environmental factors.[18 25 28 29] A recent scoping review identified several psychological and environmental barriers and facilitators of exercise in people with hip or knee OA.[23] However, scoping reviews lack the methodological rigour of systematic reviews (SRs).[30] A SR of qualitative data holds promise for a thorough and in-depth understanding of the modifiable psychosocial factors predicting PA behaviour.

The objectives of this study were to: identify, appraise and synthesise the existing qualitative evidence on barriers and facilitators to PA in hip or knee OA; explore differences in barriers and facilitators between lifestyle PA accrued in daily activities and those reported in regard to structured exercise programme specifically and between PA uptake and maintenance.

## METHODS

This SR was registered with the International Prospective Register of SRs (CRD42016030024) and its protocol reported previously.[31] The reporting follows the Preferred Reporting Items for SRs and Meta-Analyses and the Enhancing Transparency in Reporting the Synthesis of Qualitative Research statements (see online supplementary file 1).

### Population, Intervention, Comparators, Outcomes were adapted to inform eligibility
#### Population
Study participants were adults with physician diagnosed or radiographic (Kellgren and Lawrence grade ≥2) hip or knee OA or met classification criteria for OA at these joints.[32] If a study included people with other arthritis, for example, rheumatoid arthritis, they were included if people with knee or hip OA were the largest proportion. Studies with participants awaiting total joint replacement were excluded.

#### Outcomes
The perceptions of barriers and facilitators that influence uptake or maintenance of PA were the study outcomes. Studies were included if they explored the factors/barriers/facilitators/motivation to engagement in PA or addressed the experience of people with hip or knee OA regarding PA or exercise.

#### Study designs
Qualitative or mixed methods studies.

#### Language
Published in English.

### Information sources
Medline (Ovid Medline(R) in-process and other non-indexed citations and Ovid Medline(R) 1946 to present, Ovid), Embase (1974 onwards, Ovid interface), PhychINFO (1967 onwards, OVID), Web of Science, Cumulative Index to Nursing and Allied Health Literature, SPORTDiscus and Scopus were searched up to 31 of December 2015. Grey literature sources were explored, that is, OpenGrey, National Health Service evidence. The search strategy was complemented by hand search of qualitative-research-centred journals screening of references of included articles and contacting researchers active in the field.

### Search
The search strategy contained exhaustive keyword combinations for each of the four concepts of interest, that is, knee or hip OA; PA/exercise; facilitators, barriers, motivation, uptake, maintenance; qualitative studies (see online supplementary file 2).

### Study selection
The search and study selection was conducted by two researchers independently (AMK and AsA). Endnote V.X7 was used for data management. Citations and abstracts were imported and duplicates removed. After title/abstract screening, full text of potentially relevant studies were assessed and additional information was sought from authors where necessary. If consensus was not reached between the two researchers, a third reviewer was consulted (AR).

### Data collection and appraisal
All text under the sections of 'results' and 'findings' of the selected studies was considered as data items. Where findings and discussion were presented together, the whole section was considered for analysis. Data items were entered into and managed with NVivo V.11 qualitative data analysis software (QSR International).

Quality appraisal aimed to assess the reporting, methodological rigour and conceptual consistency of the included studies[33] and to identify and discard low-quality studies. Two approaches were used, which complement each other[31]: (a) the Critical Appraisal Skills ProgrammeQualitative Checklist.[34] Studies were rated as high, medium and low quality if they met ≥8, 5–7 and 4 or fewer criteria, respectively; (b) the evaluative criteria of credibility, transferability, dependability and confirmability that assess the trustworthiness of the study. Studies were rated high, medium, and low quality if they met ≥3, 2, or one and less criteria.[35] Two reviewers independently appraised the selected studies (AK and NE).

The phenomenon of interest was the description and interpretation of OA patients' perceptions and experiences regarding what facilitates, motivates or hinders them from engaging in PA. In addition, observed differences in facilitators and barriers to uptake and maintenance of PA (exercise and lifestyle PA) were also included.

## Synthesis of results

Data were analysed by thematic synthesis.[36] First, authors' interpretations and informants' quotes were coded separately, line by line. Codes of original themes, subthemes and codes clearly referring to other types of arthritis where excluded from the synthesis. Next, descriptive themes were formed through code merging and grouping in a highly iterative process, creating a hierarchical tree. To form the analytical themes, a data-driven analysis was initially conducted to allow an inductive interpretation. A group (AMK, NE, AR, JLD) review meeting was held and the fit of this synthesis within theoretical models of behaviour change, motivation, human development and health was examined. The findings showed good fit with the biopsychosocial model of health,[37] which was chosen to facilitate a more comprehensive and meaningful interpretation of the data and reporting of the findings. The descriptive themes were then re-examined and refined. At this point, the research question was introduced to help infer the barriers and facilitators under the three domains of the biopsychosocial model. To enhance the credibility of the findings, the synthesis was conducted by AMK and checked independently by NE.

## Additional analysis

The descriptive study characteristics were examined in relation to the secondary research objectives. Due to insufficient evidence, no further analysis was conducted.

## RESULTS
### Study selection

Five thousand four hundred and forty-nine studies were identified, and after removing duplicates, 2657 titles or/ and abstracts were screened and 51 full-text papers were assessed. Seven authors were contacted for further information. Information was not provided for two studies, which were excluded. Ten studies were included[38–47] (figure 1).

### Study characteristics

There were 173 participants, mainly middle aged to older, and female. Nine of 10 studies reported qualitative methodologies (table 1).

### Appraisal of studies

All selected studies were of medium or high quality (table 2). The research design and data analysis were not clear or well described in half of the studies and very few studies had clearly identified the relationship between the researcher and participants. Credibility, transferability and confirmability were met by almost all studies, although dependability only by two.

## Synthesis of results

Barriers and facilitators are presented under the three conceptual domains, that is, physical health, intrapersonal factors and social-environmental factors. Barriers and facilitators that appeared in at least three studies are reported to keep a balance between richness and applicability of the findings (table 3; see online supplementary file 3 for supporting references). When comparing exercise and PA focused studies, the themes were similar in context and equally represented in most cases. Where there are differences, these are reported.

### Physical health
#### Barriers

Physical barriers and limitations. Pain is aversive, stressful and inherent to living with OA.[38–46] It was mentioned as part of daily experience[44 45] or in relation to particular types of activities.[39–41 43 45 46] Along with fatigue and stiffness,[43–45] these symptoms hindered the ability to engage in PA. There was a vicious cycle between symptoms and lack of exercise.[40 41] At an advanced stage of OA, PA was inhibited.[41] OA symptoms were aggravated by obesity and made PA more difficult.[38 41 44] Participants also discussed their sense of limited physical capacities and that one's body cannot manage PA requirements, resulting in loss of previous activity patterns.[41–45] For example, some talked about the need to choose between activities because of limited energy.[43] Old age and lack of physical fitness were also reported as perceived PA barriers.[41 44]

#### Facilitators: PA for mobility, symptom relief and health

Among those who held a physically active lifestyle maintaining or regaining their mobility was a strong motive for PA.[38 40 44 45 47] In most cases, the aim was to keep functioning,[38 41 43 46] in some it was so specific as to prevent joint surgery.[40 47] Pain relief is another strong motive for being physically active and active individuals were more likely those who had experienced pain reduction.[38 40 44 45 47] A few informants presented a 'no pain, no maintenance' pattern, where pain cessation was followed by dropping exercise.[38 47] Improvements in other symptoms, such as stiffness and joint stability, were sufficient reasons for being active, even when pain remained.[38 44] Maintaining good general health and physical condition were also reasons for being physically active.[40 41 43 44 46] This facilitator was closely linked to a positive, beneficial PA experience and subsequent positive attitude towards PA, which is a crucial facilitator discussed below.

### Intrapersonal/psychological factors
#### Experience and beliefs about exercise

*Facilitators.* Exercise as beneficial. Experiencing benefits from exercise participation, which in most of the studies was related to engagement in an exercise intervention, helped shaping positive beliefs and motivated individuals towards continuing exercise.[38–41 44 46 47] A sense of

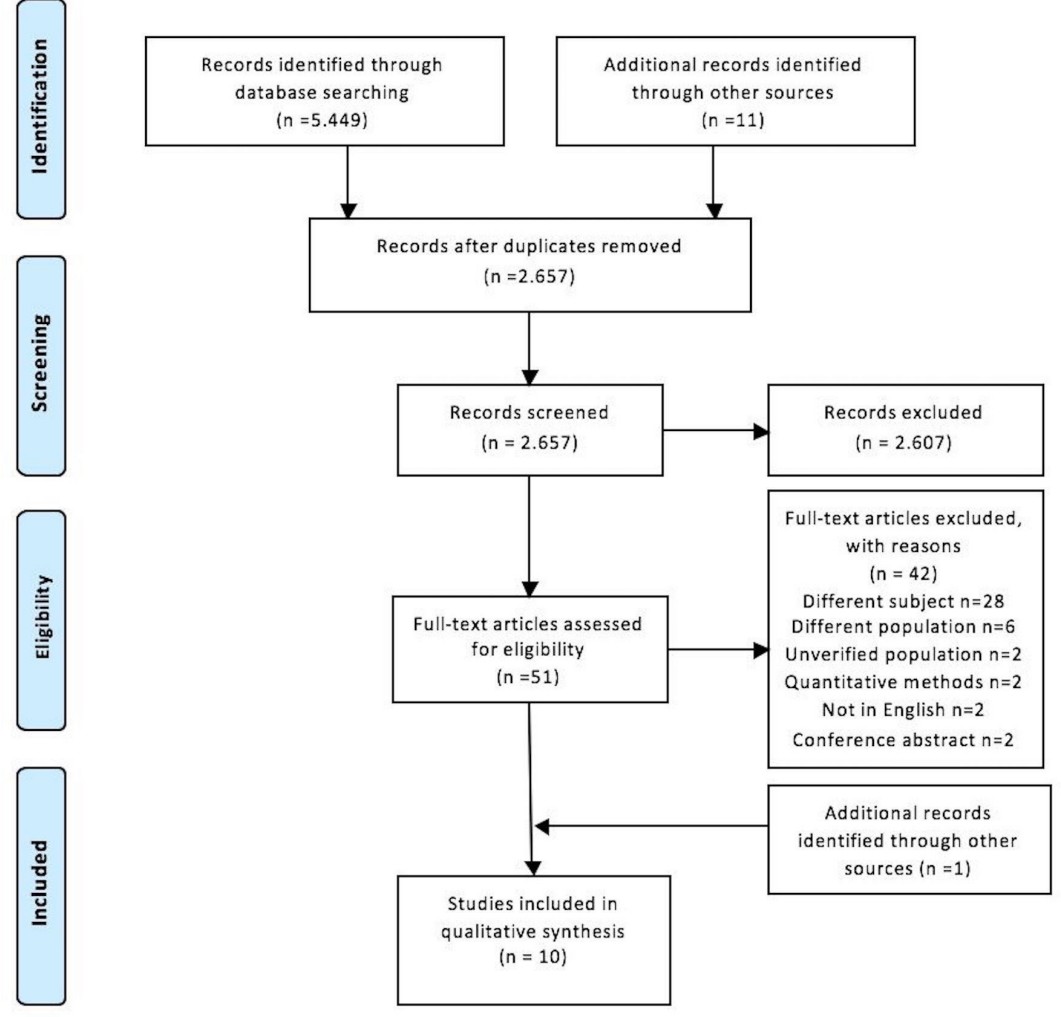

From: Moher D, Liberati A, Tetzlaff J, Altman DG, The PRISMA Group (2009). *Preferred Reporting Items for Systematic Reviews and Meta-Analyses: The PRISMA Statement. PLoS Med 6(7): e1000097. doi:10.1371/journal.pmed.1000097*

For more information, visit www.prisma-statement.org.

**Figure 1**  Study selection Preferred Reporting Items for Systematic Reviews and Meta-Analyses flow diagram.

psychosomatic well-being was an important component of this theme.[39–41 44 46] Improvement in coping with OA[46] and sleep[44] were mentioned.

Knowledge about exercise in OA. Accurate knowledge of the importance of exercise in OA, acquired through healthcare, physiotherapy and exercise interventions, was an important facilitator.[40 44–46] It led to awareness regarding exercise benefits and helped in making correct interpretations of exercise experiences.

Both the above themes emerged from exercise-focused studies only.

*Barriers.* PA as non-effective, harmful or of doubtful effectiveness. The belief that PA does not help or might further deteriorate their condition hindered people from being active.[38 40 41 43 45 46] Experiencing activity-related pain in the joint, for example, was often interpreted as PA exacerbating OA, which stemmed from the understanding

of OA as a 'wear and tear' condition.[41 43 46] Not experiencing the anticipated beneficial effects during exercise interventions was a reason for distrust in PA as an effective means of treatment.[38 40 41 46] Also, early negative experiences with sports resulted in exercise avoidance.[44]

*OA beliefs.* Beliefs that nothing can be done regarding the condition[41 44 46] and that overuse was the cause of OA[38 41 43] were linked to less inclination towards being physically active. In one study, the relationship between PA and OA was discussed as bidirectional.[43] These beliefs were mostly reported in exercise-focused studies (four exercise studies with one PA-focused study also revealing such beliefs).

*Daily activities as PA.* This theme revolved around beliefs about non-leisure PA.[41 43 44 46] However, there were no consistent patterns across studies to be clearly classified as barriers or facilitators. For example, non-leisure activities

**Table 1** Study characteristics

| Study | Objectives | Country | Participants (number; diagnosis/OA site; characteristics; sampling) | Methods (data collection and analysis) | Findings | Relevance to secondary objectives (exercise vs lifestyle PA; uptake vs maintenance) |
|---|---|---|---|---|---|---|
| Campbell et al[38] | Compliance with a physiotherapy intervention. | UK | 20 participants; Knee OA 14 female, age >45; Maximum variation sampling | Interviews; constant comparative method | Factors related to compliance: moral obligation towards the physiotherapist (initial compliance); viewing exercise as beneficial, fitting exercises in daily life, perceived symptom severity, arthritis and comorbidity attitudes, exercise and OA experiences (continued compliance) | Exercise regime Both initial and continued compliance explored |
| Fisken et al[39] | Reasons for ceasing participation in aqua-based exercise | New Zealand | 11 participants; various OA sites, 10 hip or knee; female; age >60; purposeful sampling | Focus groups; general inductive thematic approach | Main barriers: lack of appropriate classes and knowledgeable instructors, increase in pain, cold water and facilities | Exercise regime No uptake-maintenance distinction |
| Hammer et al[40] | Self-efficacy in relation to PA maintenance among maintainers and non-maintainers postintervention | Denmark | 15 participants; hip OA; 8 female, age 65–74; Criterion-based purposeful sampling | Semistructured interviews; directed content analysis | Themes: mastery experiences, vicarious experiences, verbal persuasion, physiological and emotional states, altruism | Exercise regimes No uptake-maintenance distinction |
| Hendry et al[41] | Views towards exercise, determinants of acceptability and motivation barriers | UK | 22 participants; knee OA; 16 female, age 52–86; purposeful sampling (inclusion/exclusion criteria) | Interviews and focus group; principles of framework method of qualitative analysis | Exercise participation determinants: perception of physical capacity, beliefs about exercise, motivational factors | Exercise (broad definition) No uptake maintenance distinction |
| Kabel et al[42] | Pain, social pressure and embarrassment in activity-related decision-making. | USA | 10 participants; knee OA; seven female, mean age 60; sampling method not clearly reported | Interviews; grounded theory or constant comparative method | Four PA-related patterns: Risk pain and embarrassment; risk pain, avoid embarrassment; avoid pain, risk embarrassment; avoid pain and embarrassment | PA (living with OA). No uptake maintenance distinction |
| Kaptein et al[43] | PA perception in the context of managing arthritis and multiple roles | Canada | 40 participants; 17 hip/knee OA, 16 RA, four both OA and RA, three other OA sites; 24 female, ages 29–72; purposeful sampling | Focus groups; qualitative content analysis | Positive PA perceptions, complex relationship between PA, arthritis and life roles (PA as potential cause of arthritis, reciprocal relationship, harms and benefits, perceived choices) | PA No uptake maintenance distinction |
| Petursdottir et al[44] | Exercise experience. What determines whether people exercise | Iceland | 12 participants; various OA sites, 10 hip or knee; 9 female, mean age 67 (50–81); purposeful sampling | Interviews; phenomenology (Vancouver School) | Barriers/facilitators: internal (individual attributes and exercise experiences) and external (social and physical environment) | Exercise No uptake maintenance distinction |
| Stone and Baker[45] | Facilitators and barriers to regular PA | Canada | 15 participants, hip or/and knee OA; 9 female, age 30–85; snowball sampling. | Semistructured interview; interpretational analysis | Facilitators: pain relief, clear communication from healthcare professionals, social support. Barriers: pain, psychological distress, lack of support from healthcare professionals | PA No uptake maintenance distinction |
| Thorstensson et al[46] | Underlying processes leading to response or non-response to exercise as treatment | Sweden | 16 participants; knee OA; 6 female, age 39–64; purposeful sampling | Interviews; phenomenography | Themes: to gain health, to become motivated, to experience the need for support, to experience resistance | Exercise No uptake maintenance distinction |
| Veenhof et al[47] | Factors that explain differences between patients who integrated activities in their daily lives or not | The Netherlands | 12 participants; hip or knee OA; 8 female, ages 51–80; deliberate sampling for heterogeneity | Interviews; grounded theory | Long-term goals and active involvement in the intervention related to greater adherence | Exercise No uptake maintenance distinction |

OA, osteoarthritis; PA, physical activity; RA, rheumatoid arthritis.

**Table 2** Appraisal of studies

| CASP Qualitative Checklist | Campbell et al[38] | Fisken et al[39] | Hammer et al[40] | Hendry et al[41] | Kabel et al[42] | Kaptein et al[43] | Petursdottir et al[44] | Stone & Baker[45] | Thorstensson et al[46] | Veenhof et al[47] |
|---|---|---|---|---|---|---|---|---|---|---|
|  | 6/10 | 6/10 | 6/10 | 9/10 | 6/10 | 7/10 | 9/10 | 9/10 | 7/10 | 6/10 |
| 1. Was there a clear statement of the aims of the research? | ✓ | ✓ | ✓ | ✓ | ✓ | ✓ | ✓ | ✓ | ✓ | ✓ |
| 2. Is a qualitative methodology appropriate? | ✓ | ✓ | ✓ | ✓ | ✓ | ✓ | ✓ | ✓ | ✓ | ? |
| 3. Was the research design appropriate to address the aims of the research? | ? | ✓ | ✓ | ✓ | ? | × | ✓ | ✓ | ? | ? |
| 4. Was the recruitment strategy appropriate to the aims of the research? | ✓ | ? | ✓ | ✓ | ✓ | ✓ | ✓ | ✓ | ? | ✓ |
| 5. Was the data collected in a way that addressed the research issue? | ✓ | ? | × | ✓ | ? | ✓ | ✓ | ✓ | ? | ✓ |
| 6. Has the relationship between researcher and participants been adequately considered? | ? | ? | × | ✓ | ? | × | ✓ | ? | ✓ | ? |
| 7. Have ethical issues been taken into consideration? | ? | ✓ | ✓ | ? | ✓ | ✓ | ? | ✓ | ✓ | ✓ |
| 8. Was the data analysis sufficiently rigorous? | ? | ? | ? | ✓ | ? | ? | ✓ | ✓ | ✓ | ✓ |
| 9. Is there a clear statement of findings? | ✓ | ✓ | ? | ✓ | ✓ | ✓ | ✓ | ✓ | ✓ | ✓ |
| 10. How valuable is the research? | ✓ | ✓ | ✓ | ✓ | ✓ | ✓ | ✓ | ✓ | ✓ | ? |
| Trustworthiness Credibility | ✓ | ✓ | ✓ | ✓ | ✓ | ✓ | ✓ | ✓ | ✓ | ✓ |
| Transferability | ✓ | ✓ | ✓ | ✓ | ✓ | ✓ | ✓ | ✓ | ✓ |  |
| Dependability |  |  |  |  |  |  |  | ✓ | ✓ |  |
| Confirmability | ✓ | ✓ | ✓ | ✓ | ✓ | ✓ | ✓ | ✓ | ✓ | ✓ |

✓, yes; ×, no; ?, uncertain; CASP, Critical Appraisal Skills Programme.

**Table 3** Barriers and facilitators: themes, subthemes and number of supporting references

| Domain | Major themes | Barriers | No of studies | No of references | Facilitators | No of studies | No of references |
|---|---|---|---|---|---|---|---|
| Physical health | | Physical barriers and limitations (pain and other symptoms; perceived functional limitations) | 9 | 94 | PA for mobility, symptom relief and health (PA to maintain mobility; PA for symptom relief; PA for health) | 9 | 34 |
| Intrapersonal/ psychological factors | Experience and beliefs about PA and OA | PA as non-effective, harmful or of doubtful effectiveness | 6 | 36 | Exercise as beneficial | 7 | 60 |
| | | OA beliefs | 5 | 17 | Knowledge about exercise | 3 | 8 |
| | Behavioural regulation and attitude | Resigned to OA | 5 | 10 | Keep going despite OA | 7 | 18 |
| | | Lack of motivation | 6 | 14 | Adjustments, prioritisation and personal effort (adjusting PAs; prioritising PA; personal responsibility and effort in being physically active) | 9 | 41 |
| | | Lacking behavioural regulation | 4 | 23 | | | |
| | Emotions | OA-related distress | 6 | 23 | Enjoyment | 4 | 22 |
| Social environment | Health professionals | Lack of advice and encouragement from health professionals | 5 | 22 | Support from health professionals | 8 | 50 |
| | Social support | Social comparison as demotivating | 5 | 15 | Social support facilitating PA | 7 | 43 |
| | | Lack of social support | 4 | 8 | | | |

OA, osteoarthritis; PA, physical activity.

were viewed as a sufficient amount of PA by some[41 44 46] and as insufficient by others.[41]

*Behavioural regulation and attitude*
*Facilitators.* Keep going despite OA. Authors' interpretations related to this concept varied, for example, determination to take control of arthritis,[41] perseverance,[46] personality traits of adaptability and initiative,[44] belief that there are 'things patients can do' about their OA[38] and motivation towards long-term goals.[47] The importance of keeping a positive attitude was also discussed.[43 44] In two studies, the relevant participant quotes were presented under the themes 'risking embarrassment'[42] and 'bidirectional impact between PA and arthritis'.[43]

Adjustments, prioritisation and personal effort. Physically active individuals described how they were making short or long-term modifications to their PA,[39–44] such as finding a type of exercise that was suitable for their physical abilities,[39–41 44] adjusting PA intensity to their current condition,[40 42 44] even changing their job.[43] This task of continuously adjusting PAs was quite demanding.[44] Prioritising PA and fitting it into a routine was mentioned by a

number of physically active participants and reflected the importance they assigned to PA.[38 41 46 47] Active participants also acknowledged they were the main agents in managing their condition and they were consciously making efforts to stay active.[38 41 43 46]

*Barriers.* Lack of motivation. Participants in different studies referred to a lack of motivation or goal, laziness and boredom towards exercise.[38 40 41 44 46 47] These type of barriers were reported in the exercise-focused studies only and were not further explored.

Lacking behavioural regulation. In the face of the demands of other life roles and a busy schedule, especially family related, inactive participants were not prioritising PAs.[38 41 43 46] In two studies, informants referred to not finding a PA suitable for their current condition.[39 41] In one study, low self-regulation was the reason given for not exercising regularly.[41]

Resigned to OA. In half of the studies, informants expressed a resigned attitude towards making an effort to be active.[38 41 44–46] Reflecting fatalistic beliefs about OA and feelings of helplessness, this attitude was linked to attenuated motivation for being physically active.

## Emotions

*Facilitators.* Enjoyment. Enjoying exercise in general or a particular type of exercise facilitated its continuation.[39 41 44] This facilitator of engagement emerged in the exercise-focused studies only.

*Barriers.* OA-related distress. Living with OA means adjusting to a reality of decreased physical functioning and in several cases, participants talked about this experience of giving up activities, being unable to meet life roles and daily demands as distressing or embarrassing.[38 40 42–45] Mental stress,[40] extreme unhappiness and paralysing fatigue,[44] feeling broken and mentally depressed,[45] weakness[43] were used.

## Social environment
### Health professionals

*Facilitators.* Support from health professionals. Physiotherapists exerted great influence on the patients' PA/exercise habits.[38 40 41 44 45 47] Providing instructions, education, encouragement and rapport with the patient were means of facilitating exercise. Advice and prescription by doctors was another facilitator.[41 44] Supervision during exercise was valued.[38–41 46 47] Good supervision gave participants the reassurance that what they were doing was appropriate and good for their body,[41] which they needed,[45] and motivated them to exercise.[38 40 46 47] Support from health professionals was reported in four exercise studies and one PA focused.

*Barriers.* Lack of support from health professionals. Ambiguous, no or conflicting information from doctors regarding PA was a barrier.[38 41 44 45] In one study, the instructor not having specialised OA training was the reason that lead participants to discontinue their exercise.[39]

### Social support

*Facilitators.* Social support facilitating PA. Social support as a facilitator was mainly discussed in the context of exercising in a group, as well as support from family and friends. Feeling comfortable and motivated, even inspired when exercising with people of similar physical abilities and age emerged as an advantage of PA programmes.[38–41 43 45] This was of particular importance when someone was first introduced to PA.[40] Opportunities to socialise were also an advantage of group PA.[39 41] In addition, psychological and instrumental support from family and friends emerged as an asset of physically active participants, taking the form of active encouragement, expression of interest and understanding, an exercise buddy or role model.[40 43–45] Community-based support was mentioned as PA promoting.[45] This theme stemmed equally from exercise and PA-centred studies, although the focus of the former was on group exercise and the latter on important others' support for an active lifestyle.

*Barriers.* Social comparison as demotivating. Although this concept did not explicitly appear as an authors' interpretation, it emerged from informants' quotes. Being unable to keep up with others when participating in PA was a PA barrier as it provoked feelings of embarrassment

and distress.[38–42] This barrier was reported in four exercise studies and one PA focused.

Lack of social support. The lack of social support from peers and family as a barrier was discussed in relation to lack of understanding and encouragement from the person's family and social[38 43 44] and work environment.[43]

## Physical environment

*Barriers.* The cost of exercise classes,[39 43 44] limited accessibility[44 46] and lack of availability of appropriate modes,[44 45] as well as cold weather and issues regarding safety[39] were the reported environmental barriers to PA.

## DISCUSSION

This SR is the first to synthesise the existing qualitative research on barriers and facilitators to PA in knee and hip OA. Pain and physical limitations, absence of positive PA experiences and beliefs, resigned attitude and distress due to OA, lack of behavioural regulation, lack of support from health professionals and negative social comparisons when exercising in a group were important PA barriers. Symptom relief and mobility, positive exercise experiences and beliefs, knowledge, enjoying exercise, a 'keep going' attitude, adjusting and prioritising PA and having professional and social support were important PA facilitators. Overall, the findings are consistent with known PA correlates in exercise psychology,[48] theories of behavioural change[49] and results emanating from existing SRs in general (ie, non-OA specific) populations that share common characteristics with OA patients.[50–52] Present findings also outline a unique profile of PA barriers and facilitators in lower limb OA.

Factors related to physical health, specifically pain and physical function, were the most consistently reported. This indicates that OA has a central role and impact in people's lives and experiences, which is in line with previous qualitative findings that pain discussions by people with OA differ in frequency and quality in comparison to healthy individuals.[53] Importantly, physical barriers are reported both by active and inactive people. Therefore, physical barriers alone cannot explain PA behaviour with the exception of patients at very advanced stages of OA.[54] Intrapersonal and social variables are crucial in PA behaviours reported earlier.[52]

The identified barriers and facilitators are not standalone and independent entities but manifest a complex interplay. Personal experience, knowledge and beliefs about PA, exercise and OA were interwoven concepts and formed the basis of PA behaviour. Experiencing benefits from participation in an exercise programme, which was the case in most of the included studies, shapes a positive attitude towards PA.[50 51 55–57] Accurate knowledge regarding PA, exercise and OA bolstered a positive interpretation of and predisposition towards PA experience. Viewing pain as manageable versus inevitable elicited different behaviors[58 59] and, not surprisingly, patient education is a core component of healthcare and

OA management.[60] Support from health professionals becomes crucial as they can provide rationale and motivation for PA[55] and shape the patients' health experience.[53] The above factors and available social support are not independent from, but influence motivation, attitude and behavioural regulation.

Most of the PA barriers and facilitators emerged under the psychological/intrapersonal domain and were mostly OA related. The data analysis allowed for new insights into the original studies, such as the emerging theme of OA-related distress and two distinct patterns in attitude, beliefs, motivation and behavioural regulation—one facilitating and the other hindering PA. Pain and its multifaceted impact is a source of distress in OA.[24] In turn, anxiety and depressive symptoms, which are more prevalent in people with arthritis,[61] are predictors of poorer function[62 63] and pain.[28 64–66] Still cognitive processes underlying the distinct patterns are missing, for example, what distinguishes those who, for a given level of structural disease-severity and OA-related pain, exhibit a positive attitude and behavioural regulation from those who are resigned, cope ineffectively with OA stress and lack self-regulation? Explanations involving distinguishing processes and participant characteristics might lie in theoretical frameworks of behaviour change and health, which are absent in the included studies, with one exception.[40] For example, self-efficacy, self-determination and need satisfaction are precursors of behaviour in theories which have been applied to predicting and promoting PA,[67 68] whereas sense of control is a common concept in the stress and coping literature.[69] Future research should make use of theoretical knowledge and approaches to enable targeted and more effective research and interventions.[70]

All the findings reported were grounded in the three studies that scored 'high' at both sets of quality criteria,[41 44 45] along with the seven medium quality studies, which confirms their trustworthiness. However, aspects of methodology were poorly reported or explored in the selected studies, particularly those of medium quality. A consideration of the researcher–participant relationship and employing an external auditor for the decision trail (dependability) should be used to increase confidence in the findings.

The SR findings hold implications for clinical practice. All healthcare professionals who manage people with lower limb OA have a key role in facilitating PA through their advice, attitude towards OA and decision to seek multidisciplinary input for example, from physiotherapy. Even without directed advice to increase PA, health and condition-related advice and a supportive stance from healthcare professionals can influence decisions related to PA engagement.[71] In the absence of education, people with OA tend to draw from lay and often fatalistic beliefs of PA and exercise in OA. An individual assessment of the experienced impact of pain and disability, personal attitudes and circumstances, educating about the role of PA in OA management, offering feasible yet specific PA prescription and encouragement can have an impact on the persons' PA and exercise behaviour. Pain and stress-related coping

strategies, guidance through exercise prescription and effective communication are the main components of established arthritis self-management programmes.[72] Increasing the time designated to each patient within the healthcare system could allow for such practices to take place. Counselling referral and online educational tools could also affect PA behaviour.

Based on the available qualitative evidence, it was not possible to adequately explore the secondary SR questions, an issue which has been previously reported.[52 73] Only three studies focused on lifestyle PA, which is surprising considering the paradigm shift in the health literature from exercise promotion to a combination of PA promotion and sedentary time reduction.[74] Also, only one study made the distinction between PA uptake and maintenance, despite the recognition that these two stages entail different determinants.[67 75 76] In the case of people living with OA, the factors and processes leading to uptake and maintenance of overall PA need to be further explored and understood.

This SR has applied rigorous methods and provides an in-depth and meaningful understanding of the phenomenon of interest based on the accumulated existing qualitative evidence, thus moving one step forward from existing SRs.[21 22] Gaps in the existing literature were also identified. With regards to data synthesis, coding participants' quotes and authors' interpretations separately allowed aspects of the phenomenon not captured by the original studies to come to light. During data synthesis, peer review by a multidisciplinary team took place to enhance credibility. The main reviewer's background is clinical psychology, which might be reflected in the emphasis on the 'psychological' component of PA barriers and facilitators.

There are certain limitations to this study. The majority of the included studies were exercise focused, therefore might not accurately or fully represent barriers and facilitators to lifestyle PA (of which engaging in structured exercise programme is type or form). Due to resource limitations, studies not written in English were excluded. Two relevant studies were also excluded because they were in a conference abstract form and additional data were not available.[77 78] Lastly, due to the nature of the evidence, directions of the relationships and interactions among the identified factors cannot be drawn.

In summary, there is a complex interplay among the physical, intrapersonal, psychological and socio environmental barriers and facilitators of exercise and PA that bears similarities with other chronic diseases, but also includes characteristics specific to OA. Personal experiences, beliefs, attitudes and emotions, as well as the social environment, that is, healthcare and social support, are dynamic factors shaping PA behaviour. Considering that OA becomes more prevalent with age, it is important and challenging to make sustained lifestyle changes that will have a positive impact on an individual as well as at a healthcare system level. With the aim of identifying effective practices to help people with OA become more active, future research should involve behavioural intervention studies to address the factors identified above.

## AMENDMENTS TO THE PROTOCOL

Confidence in the synthesised findings was not used due to ambiguities in the suggested process (ConQual[79]), that is, regarding transparency and satisfactory justification of the assessment outcome. However, the studies-sources of each finding were checked. The three studies scoring 'high' quality at both sets of criteria informed all themes, along with the medium quality studies.

Kappa statistic was not measured. The two researchers run the searches independently for all databases following the Medline search strategy. Because of differences in operators and options at different search engines, the number of studies differed at the stages preceding study selection. Each reviewer's full text selection stage was updated by the other researcher's findings. At this stage agreement was met for all included studies.

### Author affiliations
[1]School of Sport, Exercise and Rehabilitation Sciences, College of Life and Environmental Sciences, University of Birmingham, Birmingham, UK
[2]MRC-Arthritis Research UK Centre for Musculoskeletal Ageing Research, University of Birmingham, Birmingham, UK
[3]Centre of Precision Rehabilitation for Spinal Pain (CPR Spine), University of Birmingham, Birmingham, UK
[4]Nursing, Institute of Clinical Sciences, Medical School, University of Birmingham, Birmingham, UK
[5]Department of Health Rehabilitation Sciences, King Saud University, Riyadh, Saudi Arabia
[6]Department of Rheumatology, Dudley Group NHS Foundation Trust, Dudley, UK
[7]Academic Rheumatology Unit, School of Medicine, University of Nottingham, Nottingham, UK
[8]School of Nursing, University of Ottawa, Ottawa, Canada

**Contributors** Study concept and design: JLD, AR, AMK, RK, AbA, NE. Searches: AMK, AsA. Study appraisal: AMK, NE. Data analysis: AMK, checked by NE. Data interpretation: AMK, checked by JLD, AR, NE. Manuscript draft: AMK. Manuscript review and input: JLD, AR, AbA, NE, RK. All authors provided feedback and approved the final draft.

**Funding** This review comprises part of the research requirements of a PhD to be completed by AMK, funded by the MRC-Arthritis Research UK Centre for Musculoskeletal Ageing Research.

**Competing interests** None declared.

**Provenance and peer review** Not commissioned; externally peer reviewed.

**Data sharing statement** Qualitative synthesis level electronic data (NVivo V.11) are available upon request from the corresponding author.

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
