## [Reviewer comments · BMJ Open]

ARTICLE DETAILS

TITLE (PROVISIONAL)	Barriers and facilitators of physical activity in knee and hip osteoarthritis: a systematic review of qualitative evidence
AUTHORS	Kanavaki, Archontissa; Rushton, Alison; Efstathiou, Nikolaos; Alrushud, Asma; Klocke, Rainer; Abhishek, A; Duda, Joan

VERSION 1 - REVIEW

REVIEWER	Opeyemi Babatunde Arthritis Research UK Primary Care Centre Research Institute for Primary Care & Health Sciences Keele University
REVIEW RETURNED	24-Apr-2017

GENERAL COMMENTS	Minor comments: pg 2 line 38: "Additionally, despite the positive effects on symptoms, exercise interventions do not promote sustained behavior change" this statement is a repetition in the same paragraph. page 4 line 25-29: "Quality appraisal Quality appraisal assessed the reporting, methodological rigor and conceptual consistency of the included studies 34 to identify and discard low quality" Was quality appraisal indeed used to discard low quality studies? if yes, can the authors report how many low quality studies were discarded, as well as possible differences between the low quality studies which met eligibility criteria but later discarded and included studies? pg 7 line 10-15: Appraisal of studies "All included studies were of medium or high quality (Table 2). The research design and data analysis were not clear or well described in half of the studies and very few studies had clearly identified the relationship between the researcher and participants" Can the authors kindly clarify if the statement above refers to studies that were included in synthesis (of medium to high quality) or all the studies that initially met eligibility criteria. Major Comments It appears that quality of the included studies has not been used to inform the discussion and conclusions of the review? It may be helpful to summarise and discuss possible influence (s) of risk of bias within studies on the results and conclusions of this review. The title, main aim, and justification for this review focuses on
--

	barriers & facilitators to physical activity as opposed to exercise in OA populations. However, nearly half of included studies appear to have been conducted as part of/ or as a follow up to exercise regimes for OA. Within the analysis, narrative synthesis, and discussions, it may be helpful if authors can focus a bit more on the appraisal of evidence on barriers & facilitators to physical activity. Certainly, a comparison of the two i.e physical activity and exercise will then be a logical next step. In that way, clinicians and policy makers may be able to consider which approach might bring about sustained behavior change as well as clinically important benefits for managing OA.
--	--

REVIEWER	Maik Sliepen Universitätsklinikum Münster, Germany
REVIEW RETURNED	12-Jun-2017

GENERAL COMMENTS	I comment you on the important work that you have done with this systematic review, showing that not only physical, but also psychological and social factors influence PA in hip and knee osteoarthritis patients. I do have some remarks with the manuscript though, which I believe should be adressed:  1. As you briefly adress in the limitations, the majority of the included studies (7/10) focused on exercise rather than PA. In my opinion, I would describe exercise as a distinct category of PA. It is however, not necessarily representing PA in general. I would therefore argue that a more appropriate title might be: 'Barriers and facilitators of exercise in ... qualitative evidence'. I agree that there is value in adressing the barriers and facilitators of PA, yet I believe that you lack sufficient data for this title. 2. I believe the method section has an unnecessary multitude of headings, making it difficult to read the section 'smoothly'. For example, the sections 'Data items, data collection process, quality appraisal and phenomenon of interest' could be combined into one subheading (which could e.g. be called 'data collection and appraisal'). 3. In the method section, you report that studies including other forms of arthritis (e.g. rheumatoid) were included (page 3, line 25), as long as OA affected the majority of the study population. Although the symptoms of such diseases might be similar in some patients, the mechanisms causing these symptoms are different. I would therefore argue that more elaborate information on these studies should be presented. How many studies did include multiple forms of arthritis? Furthermore, how many patients (of the 173 included ones) have been diagnosed with a different form of arthritis? I found some mentioning of the proportion of OA patients in Table 1, but it should be presented more clearly. Depending on the amount/percentage of non-OA patients, you might want to explain why you still feel these studies can be included. 4. You report that 51 full-text papers were assessed and only 10 were included in the analysis (page 5, line 31-35). You should
--

	provide more detailed reasoning for the exclusion of the remaining 39 papers (as 2 were excluded due to the lack of information). This could, but not necessarily should, be provided as a flowchart. 5. Table 1 is rather unclear and difficult to assess, due to the quantity of text in some of the columns. I would advice to present the data in a more structured manner. 6. In Table 2, you present that the dependability of the studies is insufficient in 8/10 studies, yet you do not further elaborate on this topic during the remainder of the manuscript, which I would recommend you to do.
--	--

VERSION 1 – AUTHOR RESPONSE

Reviewer: 1

Minor comments:

- pg 2 line 38: "Additionally, despite the positive effects on symptoms, exercise interventions do not promote sustained behavior change" this statement is a repetition in the same paragraph.

Response:

Thank you for pointing this out. The repetitive sentence has been deleted.

- page 4 line 25-29: "Quality appraisal Quality appraisal assessed the reporting, methodological rigor and conceptual consistency of the included studies 34 to identify and discard low quality". Was quality appraisal indeed used to discard low quality studies? if yes, can the authors report how many low quality studies were discarded, as well as possible differences between the low quality studies which met eligibility criteria but later discarded and included studies?

Response:

We have changed the wording of the sentence to accurately reflect the aims of quality appraisal, i.e. "Quality appraisal aimed to assess the reporting, methodological rigor and conceptual consistency of the included studies and to identify and discard low quality studies" (please see page 4, lines 14-15). No low quality studies were identified. Also, we have included a figure with details of the selection process (please see Figure 1, study selection PRISMA flow diagram).

- pg 7 line 10-15: Appraisal of studies "All included studies were of medium or high quality (Table 2). The research design and data analysis were not clear or well described in half of the studies and very few studies had clearly identified the relationship between the researcher and participants". Can the authors kindly clarify if the statement above refers to studies that were included in synthesis (of medium to high quality) or all the studies that initially met eligibility criteria.

Response:

All studies that met the eligibility criteria were included in the SR. To avoid confusion, we have corrected the wording from "all included studies" to "all selected studies" (page 7, line 2) and have included the study selection flow diagram (please see Figure 1).

Major Comments

- It appears that quality of the included studies has not been used to inform the discussion and conclusions of the review? It may be helpful to summarise and discuss possible influence (s) of risk of

bias within studies on the results and conclusions of this review.

Response:

Part of the SR protocol was the assessment and reporting of confidence in the findings- the equivalent of risk of bias across studies for qualitative evidence syntheses- using the recently developed ConQual approach. Due to ambiguities in the suggested process (Munn et al., 2014), the use of this tool was not feasible (please see Supplement 4 for amendments to the protocol). An examination of the studies-sources of each theme showed that the three studies that scored “high” quality at both sets of criteria (Hendry et al., 2006; Petursdottir et al, 2010; Stone & Baker, 2015) informed all the findings (themes), along with the medium quality studies. We have added a paragraph clarifying this point and further discuss issues related to study quality. Please see page 13, paragraph 2; also, abstract> findings> lines 6-7 .

- The title, main aim, and justification for this review focuses on barriers & facilitators to physical activity as opposed to exercise in OA populations. However, nearly half of included studies appear to have been conducted as part of/ or as a follow up to exercise regimes for OA. Within the analysis, narrative synthesis, and discussions, it may be helpful if authors can focus a bit more on the appraisal of evidence on barriers & facilitators to physical activity. Certainly, a comparison of the two i.e physical activity and exercise will then be a logical next step. In that way, clinicians and policy makers may be able to consider which approach might bring about sustained behavior change as well as clinically important benefits for managing OA.

Response:

Seven of the ten studies were focused on exercise. Of the remaining three, only one study directly explored PA barriers and facilitators and two explored other PA dimensions, i.e. PA in relation to managing arthritis and multiple roles; PA in relation to pain, social pressure and embarrassment. Due to the heterogeneity in the focus and the small number of PA studies, it was deemed not possible to make the PA-exercise barriers and facilitators comparison. This information was not included in the manuscript under “additional analysis” due to word limitations but we can add if the editor agrees. However, we agree that the exercise-PA distinction should better inform our results and discussion. Hence, we compared the studies-sources of each theme in relation to the study focus. We found that in most cases exercise and PA-focused studies were equally represented. Where this is not the case, we report it under the relevant theme in the results section and modified the wording where necessary. Please see page 8, lines 2-3 and throughout results section. Also, page 14, lines 2-3.

Reviewer: 2

1. As you briefly address in the limitations, the majority of the included studies (7/10) focused on exercise rather than PA. In my opinion, I would describe these as a distinct category of PA. It is however, not necessarily representing PA in general. I would therefore argue that a more appropriate title might be: 'Barriers and facilitators of exercise in ... qualitative evidence'. I agree that there is value in addressing the barriers and facilitators of PA, yet I believe that you lack sufficient data for this title.

Response:

We agree that given the focus of the included studies, the reported barriers and facilitators not necessarily represent PA in general. At the same time, we believe that the term “PA” is in line with the initial objectives, search strategy and selection criteria of this systematic review. A shifting of the subject to “exercise” would result in exclusion of the PA-related studies (30%), and –importantly– would fail to highlight the existing gap in the literature with regards to PA barriers and facilitators in the

perspective of people living with OA.

To increase the accuracy of the terms used (i) we have made changes in the results section, corresponding tables and abstract referring to exercise rather than PA as appropriate (i.e. where the emerging themes stemmed exclusively or mostly from exercise-focused studies). (ii) emphasized this limitation more throughout the manuscript. Please see Abstract, lines 11, 13, 20; Study limitations, page 2, lines 2-3; Table 3; page 8, lines 2-3 and throughout the synthesis of results section; throughout the Discussion, e.g. page 12, lines 5, 17, 20, page 13, lines 20, 22, page 14, line 18.

2. I believe the method section has an unnecessary multitude of headings, making it difficult to read the section 'smoothly'. For example, the sections 'Data items, data collection process, quality appraisal and phenomenon of interest' could be combined into one subheading (which could e.g. be called 'data collection and appraisal').

Response:

The suggested change has been made. Please see page 4, line 10.

3. In the method section, you report that studies including other forms of arthritis (e.g. rheumatoid) were included (page 3, line 25), as long as OA affected the majority of the study population. Although the symptoms of such diseases might be similar in some patients, the mechanisms causing these symptoms are different. I would therefore argue that more elaborate information on these studies should be presented. How many studies did include multiple forms of arthritis? Furthermore, how many patients (of the 173 included ones) have been diagnosed with a different form of arthritis? I found some mentioning of the proportion of OA patients in Table 1, but it should be presented more clearly. Depending on the amount/percentage of non-OA patients, you might want to explain why you still feel these studies can be included.

Response:

Only one study included patients with other forms of arthritis. That is Kaptein et al., 2014, which included 16 participants with rheumatoid arthritis and 4 with both OA and RA. This information is now detailed in Table 1. To ensure that the review findings reflect the experience and attitudes of OA patients, we took the following actions: during data synthesis, we removed all codes/ quotes clearly relevant to IA; for all themes the codes/ quotes from Kaptein et al. (2014) were compared with the findings from the other studies and the former were in congruence with these. Therefore, we feel this study (i.e. Kaptein et al., 2014) can be included. We have not reported these details in the manuscript due to word limitations but could do so if the Editor feels this is needed.

4. You report that 51 full-text papers were assessed and only 10 were included in the analysis (page 5, line 31-35). You should provide more detailed reasoning for the exclusion of the remaining 39 papers (as 2 were excluded due to the lack of information). This could, but not necessarily should, be provided as a flowchart.

Response:

Thank you for bringing this omission to our attention. We have added a PRISMA flow diagram of the study selection process. Please see Figure 1.

5. Table 1 is rather unclear and difficult to assess, due to the quantity of text in some of the columns. I would advise to present the data in a more structured manner.

Response:

We have reduced the amount of text and columns in the table and hope it is now more clear and easier to follow and interpret. Please see Table 1.

6. In Table 2, you present that the dependability of the studies is insufficient in 8/10 studies, yet you do not further elaborate on this topic during the remainder of the manuscript, which I would recommend you to do.

Response:

We understand this point of concern.

Dependability, the equivalent of reliability for quantitative research, relates to consistency in data collection. The criterion used for the assessment in the SR was the existence of an external auditor to assess the process and product of the study (as reported in the published SR protocol). Other means for ensuring dependability have also been proposed, such as detailed reporting of the study processes (Shenton, 2004). The latter is relevant to “thick description”, which is the suggested criterion for assessing transferability. It has also been argued that “reliability is unlikely to be a demonstrable strength” in qualitative researchers’ work (Long & Johnson, 2000). We believe that the use of an external auditor is a strategy that should be incorporated in qualitative research designs more often, yet we did not feel it has significant implications for the confidence in the SR findings by itself.

Acknowledging that the discussion does not account for the quality appraisal of the studies, a paragraph has been added (p.13, paragraph 2) discussing its implications for the confidence in the SR findings.

VERSION 2 – REVIEW

REVIEWER	Dr O Babatunde Keele University, United kingdom
REVIEW RETURNED	26-Jul-2017

GENERAL COMMENTS	Previous concerns have been addressed.
--

REVIEWER	Maik Sliepen Universitätsklinikum Münster, Germany
REVIEW RETURNED	31-Jul-2017

GENERAL COMMENTS	Again, I comment you for the interesting manuscript that was handed in. I still find that table 1 contains a lot of data, which makes it slightly time-consuming to interpret. It has however, definitely improved compared to the first version and describes the important findings for the included studies. I believe you have handled the remaining comments properly and find this data worth of publication.
---